# Adsorption Characteristics of Activated Carbon for the Reclamation of Eosin Y and Indigo Carmine Colored Effluents and New Isotherm Model

**DOI:** 10.3390/molecules25246014

**Published:** 2020-12-18

**Authors:** Ratna Surya Alwi, Ramakrishnan Gopinathan, Avijit Bhowal, Chandrasekhar Garlapati

**Affiliations:** 1Department of Chemical Engineering, Fajar University, South Sulawesi 90231, Indonesia; 2Department of Chemical Engineering, Pondicherry Technological University (Formerly Known as Pondicherry Engineering College), Pondicherry 605014, India; rg_3002@yahoo.co.in (R.G.); chandrasekar@pec.edu (C.G.); 3Department of Chemical Engineering, Jadavpur University, Kolkata 700032, India; avijit_bh@yahoo.co.in

**Keywords:** acid dyes, adsorption equilibrium, kinetic parameters, new isotherm model

## Abstract

The adsorption response of eosin Y and indigo carmine acid dyes on activated carbon as a function of system temperature for a fixed concentration was investigated at various temperatures via adsorption isotherms and their thermodynamic quantities such as enthalpy, entropy, and Gibbs free energy changes. The adsorption data were exploited to develop a new adsorption isotherm. The new isotherm was developed with the spirit of solid–liquid phase equilibrium and regular solution theory. The proposed model has four adjustable constants and correlates adsorption isotherm in terms of the system temperature and melting temperature of the dye. The effect of pH on the removal of acid dyes was reported. The pH variation was observed to affect the adsorption efficiency. The removal of eosin Y and indigo carmine decreased from 99.4% to 82.6% and 92.38% to 79.48%, respectively, when the pH of the solution varied from 2 to 12. The thermodynamic analysis of the process reveals that the process of the removal of acid dyes is exothermic and spontaneous. Moreover, the kinetics parameters of the batch process are reported.

## 1. Introduction

Water pollution is the most important concern in developing and underdeveloped nations. Over the past few decades, dyes have been the major causes of water pollution [1]. The textile, paper, leather, printing, and cosmetic industries use these dyes, with the effluents released from these industries causing major contamination and resulting in the freshwater depletion. Therefore, it is imperative to preserve freshwater sources and utilize the treated water discharged from the industries for reuse [1]. Many treatment methods have been adapted, such as osmosis, membrane separations, precipitation, adsorption, and chemical oxidation, to treat the effluents [2,3,4,5,6]. The removal of pollutants by adsorption on activated carbon is a commonly used method for treating dye effluents because of the characteristics of high surface area and developed pore structures [6,7,8,9,10,11,12,13]. Adsorption data are some of the most important information for the effective implementation of adsorption process [11,12,13].

In this work, the effects of temperature and pH on the removal of eosin Y and indigo carmine dyes were investigated. Eosin Y and indigo carmine dyes are commonly known as acid dyes or anion dyes. The kinetics of these acid dye removals through adsorption process and the parameters effecting the removal of dyes [14,15,16,17,18,19] such as initial dye concentration were also investigated. Importantly, the isotherm data measured in this study were analyzed with a new adsorption isotherm model.

## 2. Results and Discussion

### 2.1. Model Correlations

The adsorption equilibrium results for eosin Y and indigo carmine acid dyes from aqueous solutions on activated carbon are reported at temperatures 303, 313, and 323 K and atmospheric pressure (*p* = 0.1 MPa). It is observed that the activated carbon uptake of dye is found to increase with an increase in temperature for both the compounds. An increase in temperature increases the solubility of the dye and hence the adsorption increases. Increased adsorption may also be as a result of the increase in the mobility of the dye ion with temperature. An increasing number of molecules may acquire sufficient energy to undergo an interaction with the active site at the surface. Therefore, an increase in the sorption uptake of dyes with an increase in temperature may be partly attributed to chemisorptions. The adsorption data established in this study are correlated with the new model and with the existing Khan et al. model. The important difference between the new model and the Khan et al. model is that it considers the influence of temperature. The new model gives an infinite dilution activity coefficient without the group contribution method, which is an interesting factor compared to other models present in the literature. The proposed model and existing model correlation effectiveness is reported in terms of the AARD% (average absolute relative deviation percentage), and the results are reported in Table 1. From Table 1, for AARD% the correlating ability of the Khan model is poor, therefore it is not presented in the figure.

From the Table 1 results, it is clear that the new model is capable of correlating adsorption data successfully. Figure 1 and Figure 2 show the correlating ability of the new model for indigo carmine and eosin Y, respectively.

### 2.2. Effect of pH

The pH of the aqueous solutions varied from 2 to 12 to study the effect of the removal of dyes at 303 K. It is clearly observed from the experimental results (Figure 3a,b) that the removal efficiency of the dye on activated carbon decreases with the increase in the pH of the solution.

The interaction of the dye molecule and activated carbon may be partially electrostatic in nature. The percentage of dye removal varied from 99.43 to 82.65 for eosin Y and from 92.38 to 79.48 for indigo carmine when the pH of the solution varied from 2 to 12. The decrease in the removal percentage at highly basic conditions may be due to the electrostatic repulsion between the deprotonated dye molecule and the negatively charged activated carbon [12].

### 2.3. Kinetic Studies

The investigation of the kinetics for the removal of dyes is important as it determines the efficiency of the removal process [15]. Kinetic studies were performed with three different initial concentrations ranging from 50 to 100 mg/L, with 0.5 g of activated carbon and a solution volume of 0.0005 m^3^ at 303 K for 120 min. Many kinetic models are available in the literature [1,7,16,20]. The removal of eosin Y and indigo carmine were examined with different kinetic models.

### 2.4. PseudoFirst Order Kinetic Model

The pseudo-first order kinetic is the simplest and oldest model and can be represented by:(1)$$\frac{dq}{dt}=k_{1}\left(q_{e}-q_{t}\right),$$
where $k_{1}$(min^−1^) is the pseudo first order rate constant, and $q_{e}$(mg/g) is the dye uptake at equilibrium. By integration, the above equation becomes [11]:(2)$$\ln\left(q_{e}-q\right)=\ln q_{e}-k_{1}t.$$

### 2.5. Pseudo Second Order Kinetic Model

The another simple model is:(3)$$\frac{dq}{dt}=k_{2}{\left(q_{e}-q_{t}\right)}^{2},$$
where $k_{2}$ is the pseudo second order rate constant, and the linear form of the above equation is:(4)$$\frac{t}{q_{t}}=\frac{1}{k_{2}\;q_{e}^{2}}+\frac{t}{q_{e}}$$

### 2.6. Intraparticle Diffusion Model

The mathematical expression for the intraparticle diffusion model is:(5)$$q_{t}=k_{p}t^{0.5}+C,$$
where $k_{p}$ is the intraparticle diffusion rate constant (mgg^−1^ min^−0.5^). Kinetics data are represented in Figure 4a,b.

The pseudo second order kinetic model is found to fit the kinetic data. The calculated parameters are shown in Table 2.

### 2.7. Thermodynamic Studies

The Langmuir isotherm model parameters for eosin Y and indigo carmine are reported in Table 3. The values of Gibbs energy, enthalpy, and entropy changes are shown in Table 4. The ΔH° and ΔG° for the adsorption process show negative values, which indicates that the process is spontaneous and exothermic [6,17,18,19,20,21]. The positive value for entropy change ΔS° indicates that the eosin Y and indigo carmine dye molecules are in a more random condition at the solid–solution interface. The decrease in the values of ΔG° with the increase in temperature indicates that the adsorption process is favored at a higher temperature [19,20,21,22,23].

## 3. Experimental Procedure

### 3.1. Materials

The activated carbon was purchased from Merck. Inc (Darmstadt, Germany). To remove the surface impurities, the activated carbon was washed several times with distilled water and then dried at 100 °C for about 72 h. Nitrogen adsorption and desorption isotherms were established (at 77 K using a Coulter model SA 3100 apparatus). Using inbuilt software, the important activated carbon physical characteristics such as specific surface area, micro pore volume, pore size distribution, and total pore volume were established and reported. From our instrument, we could calculate the Langmuir surface area, the Brunauer–Emmett–Teller (BET) surface area, the t-plot surface area, the micro pore volume, and the Barrett–Joyner–Halenda (BJH) pore size distribution. Prior to the experimental run, the sample was degassed for about 1 h at 120 °C for proper readings. Table 5 represents the chemical characteristics of the activated carbon and the mean particle size. Table 6 presents the physical characteristics of activated carbon. For chemical characterization, we have used the pH drift method and the Boehm titration method. For particle size measurement, we have used the laser scattering method. The activated carbon pore volume vs pore diameter is shown in Figure 5. The activated carbon pore area vs. pore diameter is shown in Figure 6. The activated carbon BJH pore distribution is shown in Figure 7. The adsorption and desorption hysteresis plot is shown in Figure 8.

The dyes eosin Y (CAS NO. 17372-87-1, 85%) and indigo carmine (CAS NO. 860-22-0, 98%) were purchased from Nice (Mumbai, India) and Loba Chemie (Maharashtra, India), respectively. They were used as such without any further purification. For the spectrophotometer calibration, dye solutions were prepared with known weights of dyes in distilled water. The dye samples concentrations were analyzed using a spectrophotometer Jasco UV model V-630 (Tokyo, Japan) at 517 nm and 611 nm for eosin Y and indigo carmine, respectively. Figure 9a,b and Figure 10a,b show the UV response curves and calibration graphs.

### 3.2. Batch Adsorption

The removal of two acid dyes—namely, eosin Y and indigo carmine—from aqueous solutions was studied in batch mode. Activated carbon ranging from 0.1 to 0.6 g was taken in a 250 mL stoppered conical flask, and fifty milliliters of aqueous dye solutions of a known initial concentration (i.e., the initial concentration of eosin Y is 368 mg/L and of indigo carmine is 536 mg/L) was added and shaken for three days in a temperature-controlled water bath (at 303, 313, and 323 K). The equilibrium concentrations of the dyes in the solution were accurately measured with a UV (ultraviolet light) spectrophotometer. The removal of the acid dyes from the aqueous solution and the equilibrium uptake to activated carbon was calculated using the following equations.
(6)$$\%\;\mathrm{dye}\;\mathrm{removal}\;=\;\frac{C_{o}-C_{e}}{C_{o}}\times 100,$$
(7)$$q_{e}=\frac{\left(C_{o}-C_{e}\right)v}{w},$$
where $C_{o}$ is the initial concentration of dye in aqueous solution (mg/L), $C_{e}$ is the equilibrium dye concentration (mg/L), *v* is the volume of the solution in m^3^, and *w* is the weight of the activated carbon in g. The experimental results (Table 7) show the effect of temperature for the removal of eosin Y and indigo carmine from aqueous solutions on activated carbon at 303, 313, and 323 K at atmospheric pressure (0.1 MPa).

### 3.3. Scanning Electron Micrograph (SEM) of Activated Carbon

A high-performance scanning electron microscope (JSM 6390 mole, Peabody, MA, USA) with a high resolution was employed to study the surface morphology. The micrograph of activated carbon is shown in Figure 11a. The rough surface observed in Figure 11a indicates that it is suitable for the adsorption of dye from the liquid phase. To indicate this, the SEM of indigo carmine adsorbed on activated carbon is shown in Figure 11b. It is clearly evident that the adsorption of indigo carmine is uniform over the micro porous surface area in the activated carbon.

### 3.4. Thermodynamic Parameters

Gibbs energy change (ΔG°), enthalpy change (ΔH°), and entropy change (ΔS°) were estimated with the help of Langmuir isotherm parameters. More details about this method can be found in the literature [6,20,21]. The relevant equations for estimating the thermodynamic properties are listed below.
(8)$$\Delta G^{{}^{\circ}}=-RT\ln B,$$
(9)$$\Delta H^{{}^{\circ}}=-R\left(\frac{T_{2}T_{1}}{T_{2}-T_{1}}\right)\ln\left(\frac{B_{2}}{B_{1}}\right),$$
(10)$$\Delta S^{{}^{\circ}}=\frac{\Delta H^{{}^{\circ}}-\Delta G^{{}^{\circ}}}{T},$$
where *B*, *B*_1_, *B*_2_, are the Langmuir constants at T, $T_{1}$, and $T_{2}$ temperatures, respectively.

## 4. New Thermodynamic Model

In the recent past, a thermodynamic model was proposed to describe solid–fluid phase equilibrium. However, the model requires the group contribution method to evaluate their parameters. When we tried to apply those models to the solute such as indigo carmine and eosin Y, we could not succeed. Due to this, the model applicability is limited. Therefore, there is a need to develop new model to address this problem. Hence, we proposed the current model. In the proposed model, we assume that the dye molecules are uniformly absorbed on the activated carbon. The solid phase is treated as a solid solution of dye well distributed in the activated carbon [1,20,21,22,23]. At equilibrium conditions, the dye compound fugacity in the solid phase is equal to that of the dye present in the liquid phase. Equilibrium conditions can be mathematically written as:(11)$${\hat{f}}_{dye}^{S}={\hat{f}}_{dye}^{L},$$
where the superscript *S* is the solid phase and *L* is the liquid phase.

Equation (11) is rewritten in terms of the activity coefficient and the pure dye component fugacity as:(12)$$z\gamma_{dye}^{S}f_{dye}^{oS}=x\gamma_{dye}^{L}f_{dye}^{oL},$$
where x and z are the mole fraction of the dye in the liquid and solid phase; $\gamma_{dye}^{L}$ and $\gamma_{dye}^{S}$ are the activity coefficients of the liquid and solid phases; $f_{dye}^{OL}$ and $f_{dye}^{OS}$ are the standard state fugacities of the dye in their respective pure states.

The expression for $\gamma_{dye}^{L}$ may be obtained from regular solution theory [22].
(13)$$\gamma_{dye}^{L}=\exp\left(-W{\left(1-x\right)}^{2}/RT\right).$$

In Equation (13), W is the adjustable parameter. Further, W is indicative of the molecular interaction of dissimilar molecules in the solution.

The expression for $\gamma_{dye}^{S}$ may be obtained from the Redlich–Kister expansion as:(14)$$\gamma_{dye}^{S}=\exp\left({\left(1-z\right)}^{2}\left(A+Bz+Cz^{2}\right)\right),$$
where *A*, *B*, and *C* are temperature-dependent parameters.

Combining Equations (13) and (14) results in:(15)$$x=z\left[\frac{\exp\left({\left(1-z\right)}^{2}\left(A+Bz+Cz^{2}\right)\right)}{\exp\left(-W{\left(1-x\right)}^{2}/RT\right)}\right]\left(\frac{f_{dye}^{oS}}{f_{dye}^{oL}}\right).$$

We can express the pure component fugacity ratio in terms of melting temperature as:(16)fdyeoSfdyeoL=exp−6.541−TmT.

Combining Equations (15) and (16) gives:(17)x=zexp(1−z)2A+Bz+Cz2exp−W(1−x)2/RTexp−6.541−TmT.

In Equation (17), W, A, B, and C are adjustable parameters, therefore it is a4 parameter model and requires only equilibrium data at the temperature and melting point of the solute. A somewhat similar equation was proposed earlier [20]; however, the main difference between the earlier model and the present model lies in the estimation of the liquid phase activity coefficient. The earlier model [20] makes use of group contribution method to compute the activity coefficient, whereas the present method is free from any group contribution method. Due to presence of sodium in eosin Y and indigo carmine compounds, it is not possible to apply the group contribution method to evaluate the melting enthalpy and activity coefficient. However, the present model does not require any group contribution method in evaluating the activity coefficient. Therefore, this model may be used for any compound.

### Khan Model

This is the first phase equilibrium model and it is based on the phase equilibrium criteria. This model inspired us to develop a few other models in the recent past [20,21,22,23]; their applicability, success, and other limitations can be seen in recent literature [20,21,22,23]. In the present work, the Khan et al. model has been considered for comparison purposes. The Khan et al. model is:(18)$$x=z\exp\lfloor {\left(1-z\right)}^{2}\left(a+bz+cz^{0.5}\right)\rfloor ,$$
where *a*, *b*, and *c* are adjustable temperature-dependent model parameters. x and z are the mole fraction of the dye in liquid and solid phases, respectively.

## 5. Conclusions

The adsorption of eosin Y and indigo carmine from aqueous solutions on activated carbon (at 303,313, and 323 K) in atmospheric conditions (0.1 MPa) was determined. The thermodynamic model proposed in this study is successful in correlating the dye-activated carbon systems. The overall deviation between the experimental and the new model results is less than 13.05%. Thermodynamic parameters, such as enthalpy change, entropy change, and Gibbs energy change, were determined. The effect of pH and the adsorption kinetics was reported.

## Figures and Tables

**Figure 1 molecules-25-06014-f001:**
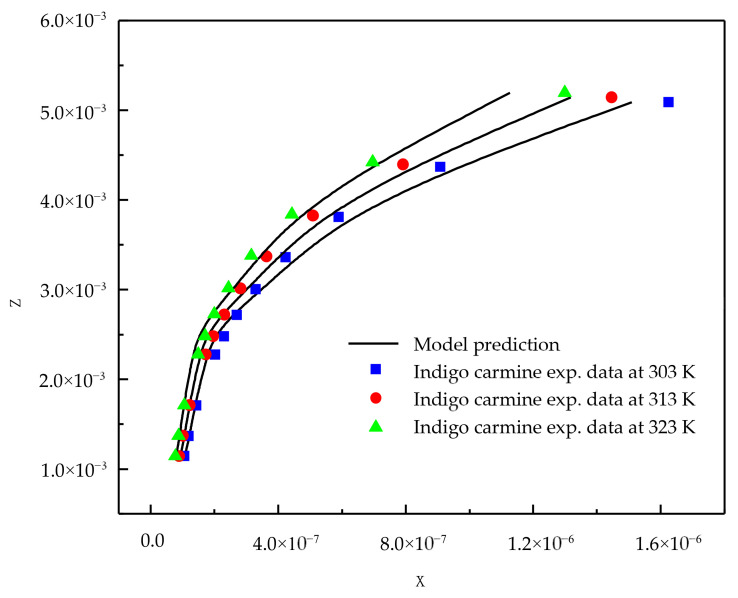
New model correlations for indigo carmine dye-activated carbon system at various temperatures.

**Figure 2 molecules-25-06014-f002:**
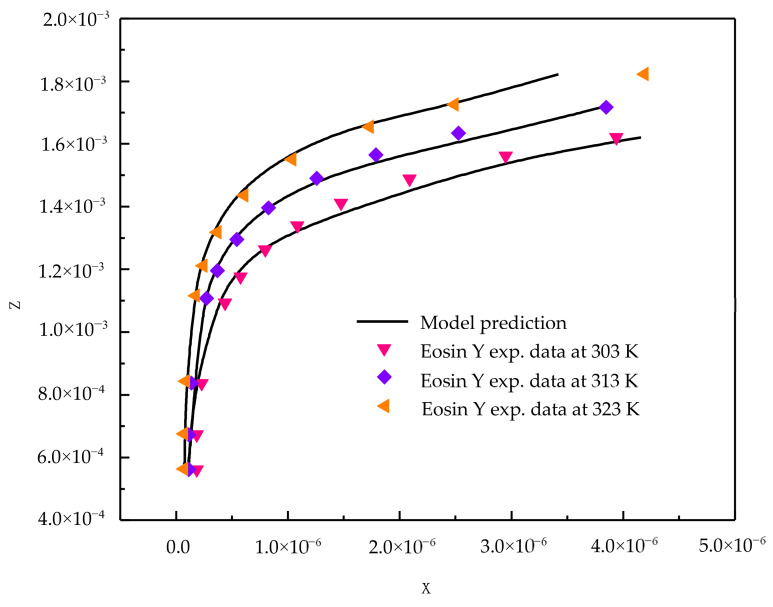
New model correlations for eosin Y dye-activated carbon system at various temperatures.

**Figure 3 molecules-25-06014-f003:**
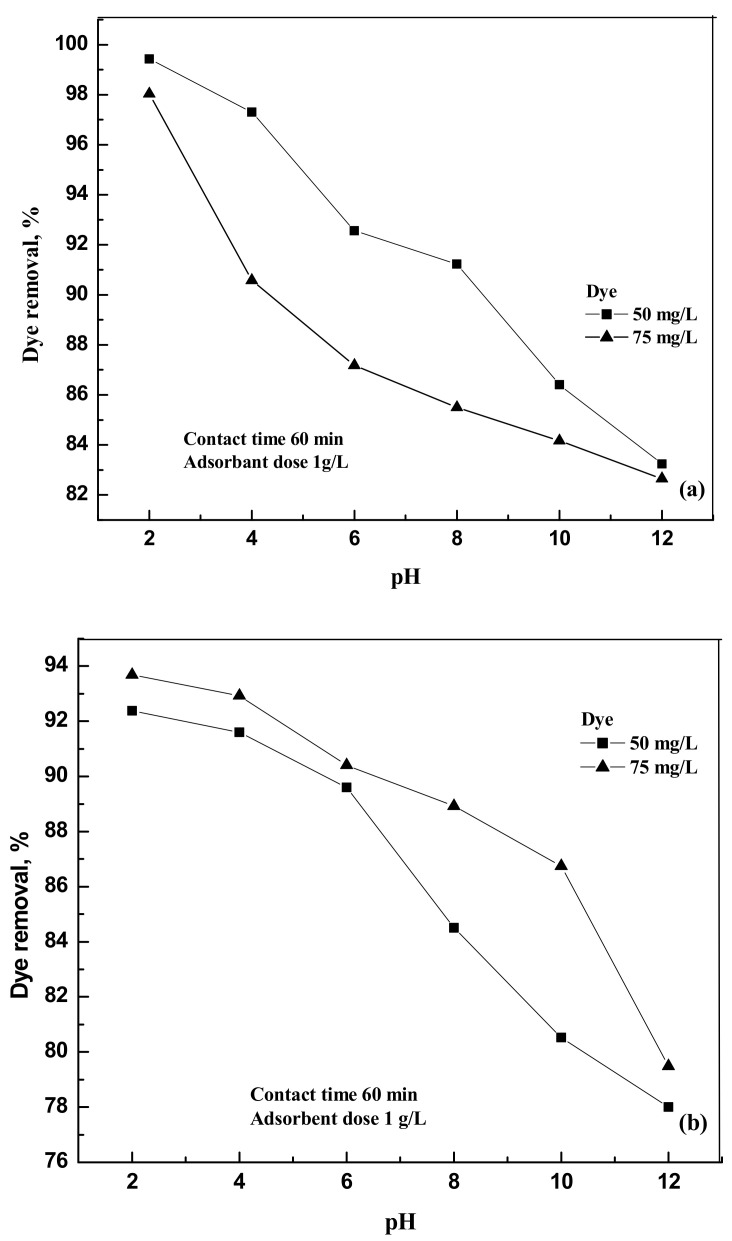
(**a**) Effect of pH on the removal of the dye compound by activated carbon at 303 K foreosin Y. (**b**) Effect of pH on the removal of the dye compound by activated carbon at 303 K indigo carmine.

**Figure 4 molecules-25-06014-f004:**
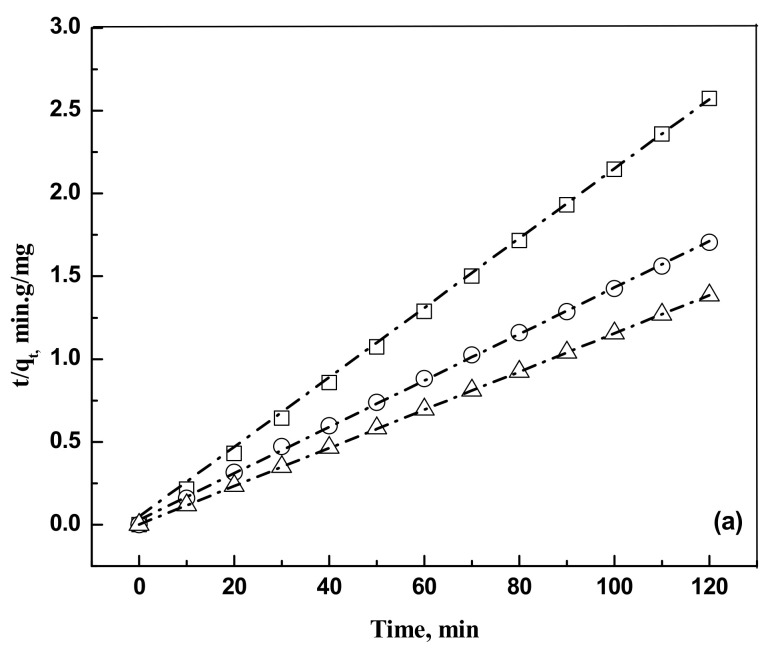
(**a**) Plot of pseudo second order kinetic modeling for the removal of eosin Y dye on activated carbon for an adsorbent dose of 0.5 g and a solution volume of 5 × 10^−4^ m^3^. Temperature, 303 K. Concentration: □, 50 mg/L; ○, 75 mg/L; and Δ, 100 mg/L. The dash dot lines are model predictions at the respective concentrations. (**b**) Plot of pseudo second order kinetic modeling for the removal of indigo carmine dye on activated carbon for an adsorbent dose of 0.5 g and a solution volume of 5 × 10^−4^ m^3^. Temperature, 303 K. Concentration: □, 50 mg/L; ○, 75 mg/L; and Δ, 100 mg/L. The dash dot lines are model predictions at the respective concentrations.

**Figure 5 molecules-25-06014-f005:**
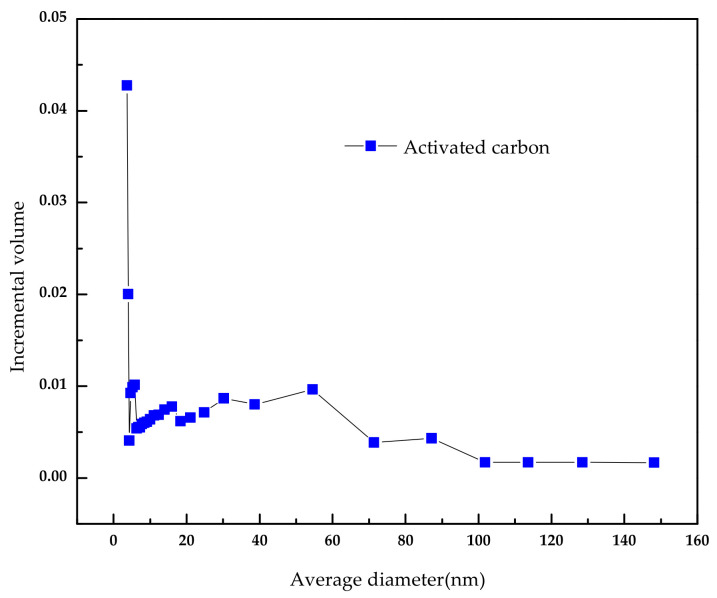
Activated carbonpore volume vs. average diameter.

**Figure 6 molecules-25-06014-f006:**
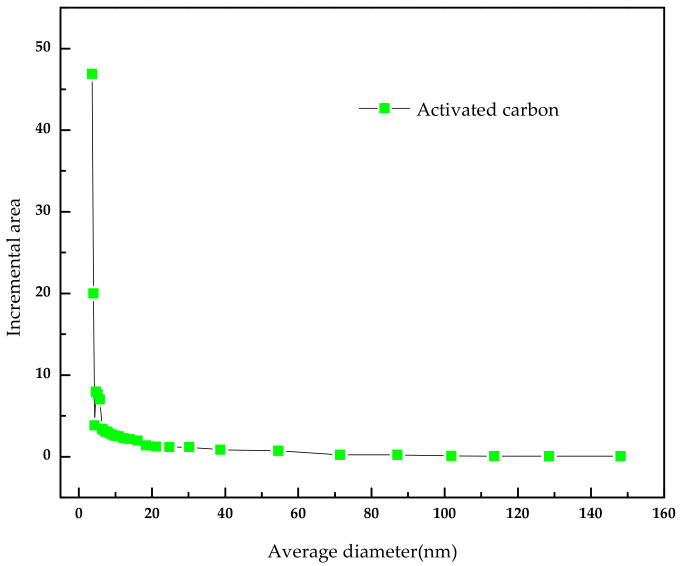
Activated carbonpore area vs. average diameter.

**Figure 7 molecules-25-06014-f007:**
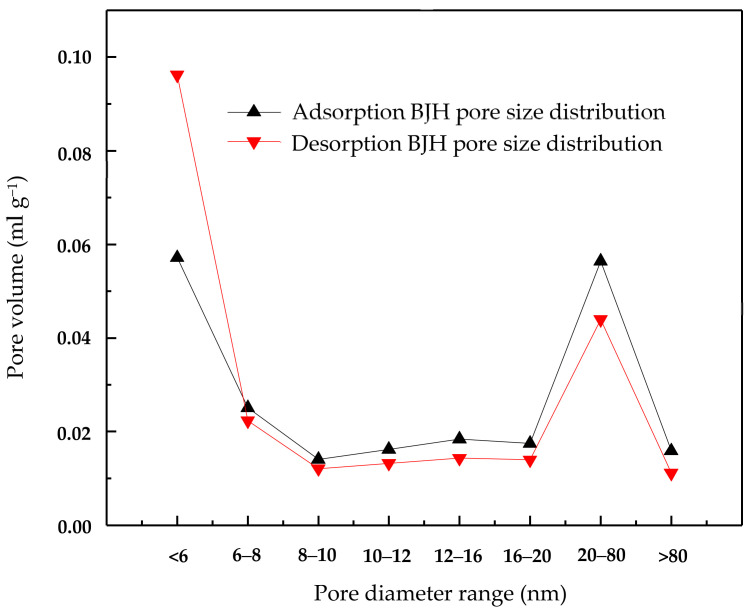
Activated carbon adsorption and desorption BJH (Barrett–Joyner–Halenda) pore size distribution.

**Figure 8 molecules-25-06014-f008:**
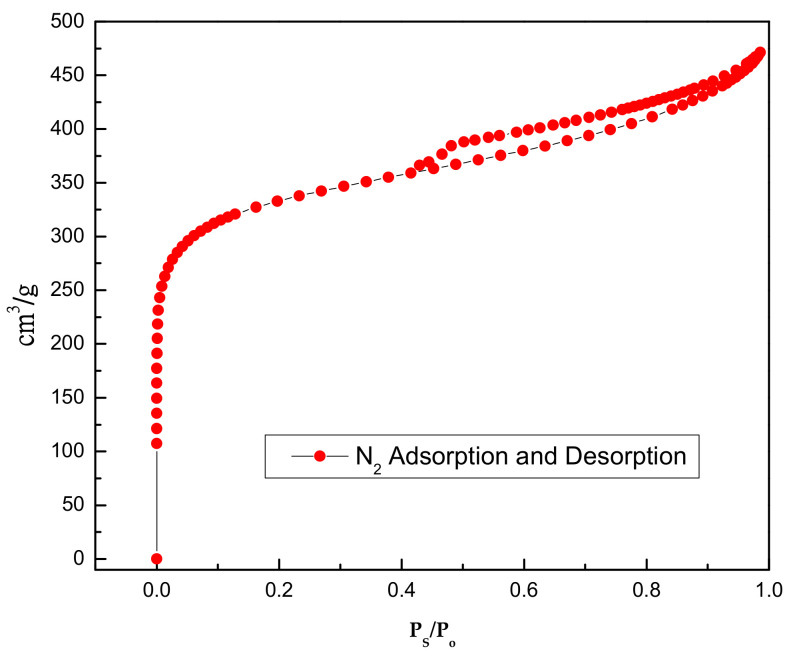
Activated carbon BET hysteresis plot.

**Figure 9 molecules-25-06014-f009:**
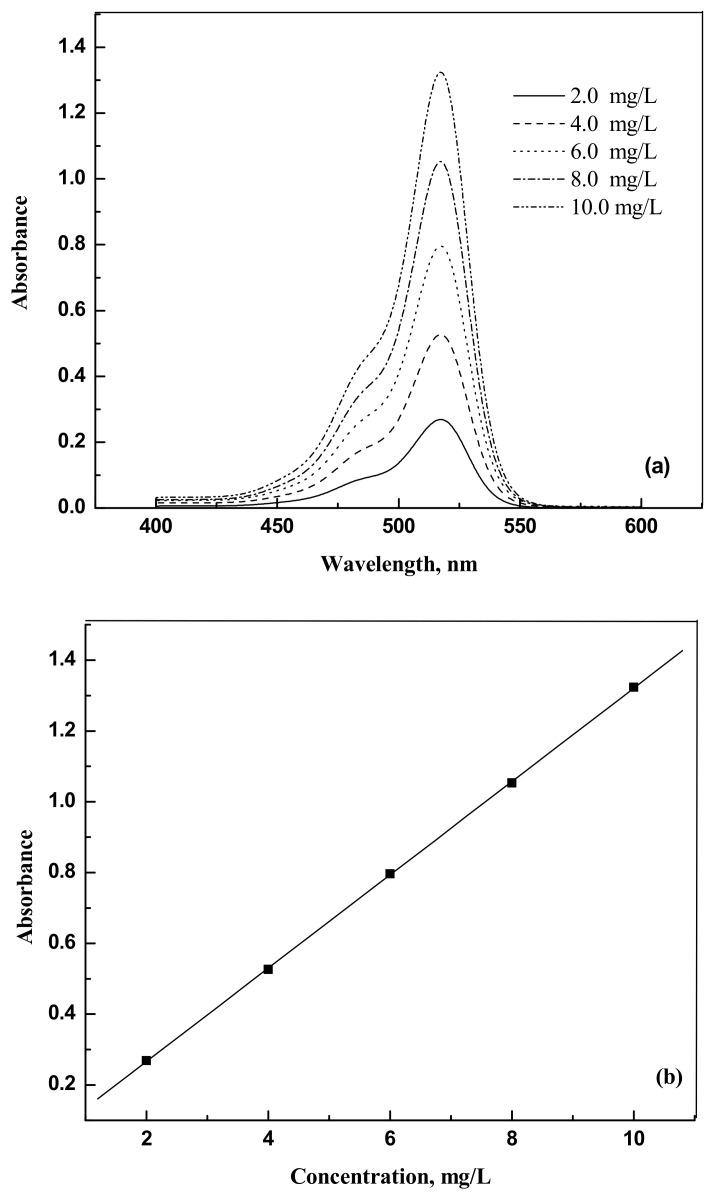
(**a**) Absorbance versus wavelength for eosin Y. (**b**) Absorbance versus concentration for eosin Y.

**Figure 10 molecules-25-06014-f010:**
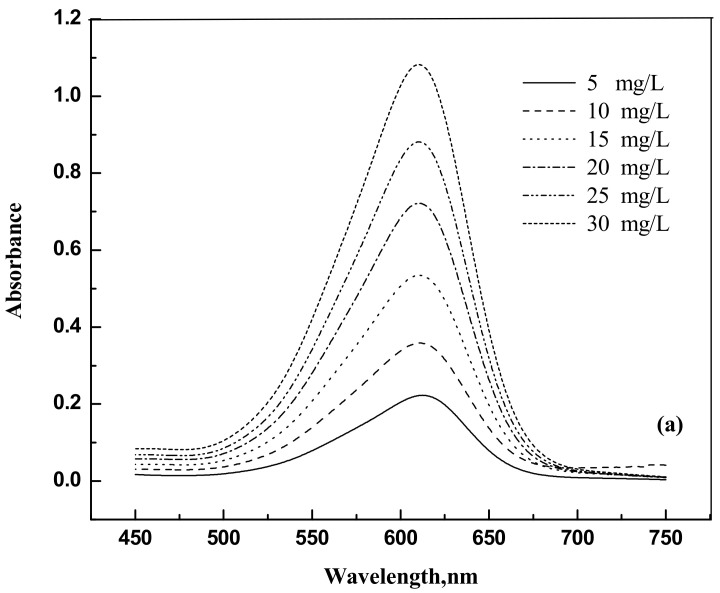
(**a**) Absorbance versus wavelength for indigo carmine (**b**) Absorbance versus concentration for indigo carmine.

**Figure 11 molecules-25-06014-f011:**
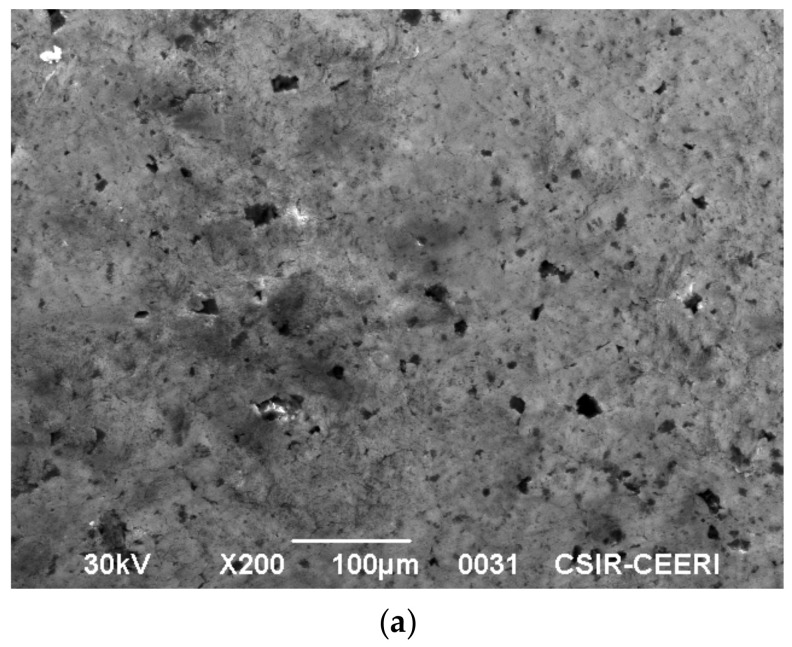
(**a**) Scanning electron microscope image of activated carbon. (**b**) Scanning electron microscope image of indigo carmine adsorbed on activated carbon.

**Table 1 molecules-25-06014-t001:** Summary of model correlations along with AARD%.

Compound	Correlations Based on Khan et al., Model				AARD%
	a	b	c		
Indigo carmine	−6.8945	1504.5	−126.50		13.05
Eosin Y	−1.8805	10599	−537.12		51
	**Correlations based on new model**				
	W	A	B	C	
Indigo carmine	−23503	−5.2068	1503.9	−126.26	6.84
Eosin Y	9508.5	−6.9458	14023	−778.97	13.7

**Table 2 molecules-25-06014-t002:** Summary of the second order rate constant values.

Dye	Concentration (mg/L)	$q_{e\left(\exp\right)}\;(\mathrm{mg}/g)$	$q_{e\left(cal\right)}\;(\mathrm{mg}/g)$	$k_{2}\;(g/\mathrm{mg}\cdot \min)$	R^2^
Indigo Carmine	50	47.02	47.28	0.02405	0.999
	75	66.89	66.93	0.72234	0.999
	100	82.81	82.37	0.02139	0.999
Eosin Y	50	46.62	47.62	0.00886	0.999
	75	70.27	71.29	0.006717	0.999
	100	86.61	86.78	0.06135	0.999

**Table 3 molecules-25-06014-t003:** Summary of the Langmuir correlation constants.

Compound	T/K	$K_{L}$ $\left(L\cdot {\mathrm{mg}}^{-1}\right)$	$q_{m}$ $\left(\mathrm{mg}\cdot g^{-1}\right)$	R^2^
Indigo Carmine				
	303	0.0998	246.98	0.985
	313	0.1127	250.16	0.985
	323	0.1300	253.31	0.984
Eosin Y				
	303	0.1206	90.57	0.998
	313	0.1367	95.99	0.999
	323	0.1914	100.88	0.999

**Table 4 molecules-25-06014-t004:** Summary of the thermodynamic parameters.

Dye	*T*/K	ΔGoJmol	ΔHοJmol	ΔSοJmol K
Indigo Carmine	303	−29,012.7	−12,010	56.09
	313	−30,286.6		58.37
	323	−31,637.3		60.74
Eosin Y	303	−29,485.5	−28,460.5	3.38
	313	−30,785.4		7.42
	323	−32,677.3		13.05

**Table 5 molecules-25-06014-t005:** Summary of the chemical characteristics of the activated carbon and mean particle size.

Compound	Point of Zero Charge ^a^	Basicity ^b^	Acidity ^c^	Particle Mean Size ^d^
Activated carbon Merck Grade	6.7	0.48	0.35	26.514 µm

^a^ pH drift method; ^b,c^ Boehm titration method, units mmol/g; ^d^ HORIBA Laser Scattering Particle Size Distribution Analyzer LA-960.

**Table 6 molecules-25-06014-t006:** Summary of the physical characteristics of the activated carbon.

Compound	^a^ Surface Area	Langmuir Surface Area (m^2^ g^−1^)	^b^ V_Total_ (cm^3^ g^−1^)	^c^ V_micro_ (cm^3^ g^−1^)	^d^ V_meso_ (cm^3^ g^−1^)	^e^ D_ave_ (nm)
	BET (m^2^ g^−1^)					
Activated Carbon	1157	1323	0.72	0.33	0.39	2.49

^a^ BET (Brunauer–Emmett–Teller) surface area; ^b^ Total pore volume, measured at P/P0 = 0.9814; ^c^ Micropore volume, based on t-plot report; ^d^ Mesopore volume, based on the difference between the V_Total_ and V_micro_ volumes; ^e^ Average pore diameter of absorbents, calculated by 4V_Total_/SBET.

**Table 7 molecules-25-06014-t007:** Equilibrium isotherm data for the removal of eosin Y and indigo carmine dye on activated carbon.

Dye	*T*/K = 303		*T*/K = 313		*T*/K = 323	
	Cemgl	qemgg	Cemgl	qemgg	Cemgl	qemgg
Eosin Y	148.9	87.64	135.9	92.84	121.6	98.58
	114.7	84.45	102.8	88.39	88.07	93.31
	86.28	80.49	71.94	84.59	54.76	89.50
	62.71	76.32	45.75	80.56	32.53	83.87
	41.98	72.45	28.31	75.49	18.60	77.65
	26.53	68.29	17.86	70.03	11.66	71.27
	18.35	63.57	12.46	64.64	7.731	65.50
	13.32	59.11	8.994	59.83	6.001	60.33
	6.378	45.20	6.007	45.25	3.348	45.58
	4.717	36.37	4.988	36.30	2.883	36.51
	4.013	30.33	3.618	30.37	2.680	30.44
Indigo Carmine						
	38.91	198.8	33.83	200.9	28.82	202.9
	24.53	170.5	21.33	171.6	18.17	172.6
	16.10	148.5	14.00	149.1	11.92	149.7
	11.841	131.0	10.30	131.4	8.771	131.8
	9.001	117.1	7.828	117.4	6.668	117.6
	6.573	105.9	5.716	106.1	4.869	106.2
	5.195	96.51	4.518	96.63	3.848	96.76
	4.556	88.57	3.962	88.67	3.375	88.77
	3.569	66.55	3.103	66.61	2.644	66.67
	3.140	53.29	2.730	53.33	2.326	53.37
	2.751	44.44	2.392	44.47	2.038	44.50

The molar densities of the water at *T* = (303, 313, and 323) K were 55,252.33, 55,061.80 and 54,830.02 $\mathrm{mol}\cdot m^{-3}$, respectively Standard uncertainties u are as follows: u(T) = ±0.1 K, u(p) = ±0.0013 MPa; the relative standard uncertainties $u_{r}$ in $u_{r}$($m_{e}$) = 0.03, $u_{r}$($q_{e}$) = 0.03.

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
