# Peer review of "Adsorption Characteristics of Activated Carbon for the Reclamation of Eosin Y and Indigo Carmine Colored Effluents and New Isotherm Model"

_molecules, 2020, doi:10.3390/molecules25246014_

Round 1

Reviewer 1 Report

It is an excellent research work and with interesting contributions to the state of the art in the area of ​​research on adsorption in aqueous phase.
Before accepting this work for publication, I suggest that the authors introduce the following changes:

1. The nitrogen adsorption-desorption isotherm should be better analyzed. I suggest to the authors for them to represent the relative pressure on a semi-logarithmic scale; with this they will be able to better analyze the area of ​​emicroporosity.
2. Additionally, they should have a broader discussion on the results of the adsorbent surface and its pore distribution.
3. I recommend using the different models like DFt, NLDFT, QSDFt and DR and DA. Realzia run comparative analysis.
4. SEM analysis needs improvement. Be clearer.
5. For me the most important and novel part of the writing is "New thermodynamic model" and it must be improved substantially, so that the reader has clarity of what is really new.
6. They should better connect the Khan model.

Author Response

Reviewer #1:

Recommendation: Publish after major revisions noted.

We thank the reviewer for recommending publication.

Comments:

1.The nitrogen adsorption-desorption isotherm should be better analyzed. I suggest to the authors for them to represent the relative pressure on a semi-logarithmic scale; with this they will be able to better analyze the area of ​​emicroporosity.

As suggested by the reviewer the Nitrogen adsorption-desorption isotherm has been analyzed using the inbuilt software. The following is included in the manuscript.

Nitrogen adsorption and desorption isotherms were established (at 77 K using Coulter model SA 3100 apparatus).Using inbuilt software the important activated carbon physical characteristics like  specific surface area, micro pore volume, pore size distribution and total pore volumes are reported. Form our instrument we could calculate Langmuir surface area, Brunauer-Emmett-Teller (BET) surface area, t-plot surface area, Micro pore volume and Barrett-Joyner-Halenda (BJH) pore size distribution. Prior to experimental run sample was degassed for about 1 hr at 120oC for proper readings.

Surface Area Report: Langmuir Surface area     1322.76 sq.m/g Correlation coefficient    0.9997

Surface Area Report: BET Surface area     1155.20 sq.m/g Correlation Coefficient     0.99932

t-Plot Surface Area Report:

t-Plot Surface Area

407.161

sq.m/g

Micropore Surface Area

748.038

sq.m/g

Micropore Volume

0.33615

ml/g

Correlation Coefficient

0.99488

Total Pore Volume    0.7233 ml/g (Ps/Po =            0.9814,   Adsorption)

  1. Additionally, they should have a broader discussion on the results of the adsorbent surface and its pore distribution.

As suggested by the reviewer we have given additional discussion on adsorbent surface and pore distribution. The following is included in the manuscript.

Table 1b. Summary of physical characteristics of activated carbon

Compound

aSurface area

BET(m2.g-1)

Langmuir Surface area(m2.g-1)

bVTotal(cm3.g‑1)

‑cVmicro(cm3.g-1)

dVmeso(cm3.g-1)

eDave(nm)

Activated Carbon

1157                   

1323

  0.72             

     0.33                                                                               

    0.39                                                                               

 2.49                                                                                      

a BET (Brunauer-Emmett-Teller) surface area;b Total pore volume, measured at P/P0=0.9814;

c Micropore volume, based on t-plot report;d Mesopore volume, based on the difference between VTotal and Vmicro volumes;eAverage pore diameter of absorbents, calculated by 4VTotal/SBET

Figure 1. Activated carbon pore volume vs average diameter

Figure 2. Activated carbon pore area vs average diameter

Figure 3. Activated carbon adsorption and desorption BJH pore size distribution  

Figure 4.Activated Carbon BET hysteresis plot.

  1. I recommend using the different models like DFt, NLDFT, QSDFt and DR and DA. Realzia run comparative analysis.

As suggested by the reviewer we have given additional discussion

  1. SEM analysis needs improvement. Be clearer.

As suggested by the reviewer we have given additional discussion
5. For me the most important and novel part of the writing is "New thermodynamic model" and it must be improved substantially, so that the reader has clarity of what is really new.

As suggested by the reviewer we have given additional discussion

  1. They should better connect the Khan model.

As suggested by the reviewer we have given additional discussion and this model is taken to compare the new model and results are indicated table 3.

Reviewer 2 Report

Please refer to the following comments:
1.The increase in dye removal at higher temperature is surprising.
This can be explained by the phenomenon of chemisorption but the
authors repeatedly emphasize the fact of adsorption is based
on electrostatic attraction. Physical (equilibrium) adsorption is
always more effective at a lower temperature. Please, provide an
exhaustive comment of such difference between fundamental knowledge
and experimental results.
2. What was the reason to use just such (
368 mg/l and 536 mg/g)
initial concentration of dyes?
3.
Many values ​​are specified with too much precision (eg BET
surface area = 1155.20 m2/g). Please round them to the values that
matches the measurement uncertainty.

Author Response

Reviewer:2

Recommendation: Publish after major revisions noted.

We thank the reviewer for recommending publication.

1.The increase in dye removal at higher temperature is surprising. This can be explained by the phenomenon of chemisorption but the authors repeatedly emphasize the fact of adsorption is based on electrostatic attraction. Physical (equilibrium) adsorption is always more effective at a lower temperature. Please, provide an exhaustive comment of such difference between fundamental knowledge and experimental results.

Dye adsorption influenced by several factors includes adsorbent dose, initial adsorbate concentration, contact time, temperature, pH and ionic strength [1-6]. Among them, pH and ionic strength are two important factors affecting the adsorption processes. Dyes are adsorbed mainly through hydrophobic and electrostatic attractions, hydrogen bonding and surface function group interactions between the adsorbents and the dyes [7-11]. Solution pH affects not only the surface charge of the adsorbents, but also the ionization of the dyes [1]. Solution pH affects the adsorption mainly through adjusting the electrostatic interactions between the dyes and the adsorbents [6-12].

  1. S. Hosseini, M.A. Khan, M.R. Malekbala, W. Cheah, T.S.Y. Choong, Carbon coated monolith, a mesoporous material for the removal of methyl orange from aqueous phase: adsorption and desorption studies, Chem. Eng. J. 171 (2011) 1124–1131.
  2. Zhou, W. Gong, C. Xie, D. Yang, X. Ling, X. Yuan, S. Chen, X. Liu, Removal of neutral red from aqueous solution by adsorption on spent cottonseed hull substrate, J. Hazard. Mater. 185 (2011) 502–506
  3. Yao, B. He, F. Xu, X. Chen, Equilibrium and kinetic studies of methyl orange adsorption on multiwalled carbon nanotubes, Chem. Eng. J. 170 (2011) 82–89
  4. K. Gupta, D. Pathania, S. Sharma, S. Agarwal, P. Singh, Remediation and recovery of methyl orange from aqueous solution onto acrylic acid grafted Ficus carica fiber: isotherms, kinetics and thermodynamics, J. Mol. Liq. 177 (2013) 325–334.
  5. Zhao, F. Zhou, L. Li, M. Cao, D. Zuo, H. Liu, Removal of anionic dyes from aqueous solutions by adsorption of chitosan-based semi-IPN hydrogel composites, Compos. Part B – Eng. 43 (2012) 1570–1578.
  6. Han, P. Han, Z. Cai, Z. Zhao, M. Tang, Kinetics and isotherms of neutral red adsorption on peanut husk, J. Environ. Sci. – China 20 (2008) 1035–1041
  7. You, Z. Wu, T. Kim, K. Lee, Kinetics and thermodynamics of bromophenol blue adsorption by a mesoporous hybrid gel derived from tetraethoxysilane and bis(trimethoxysilyl)hexane, J. Colloid. Interf. Sci. 300 (2006) 526–535.
  8. Wu, L. You, H. Xiang, Y. Jiang, Comparison of dye adsorption by mesoporous hybrid gels: understanding the interactions between dyes and gel surfaces, J. Colloid. Interf. Sci. 303 (2006) 346–352.
  9. Wu, J. Wu, H. Xiang, M. Chun, K. Lee, Organosilane-functionalized Fe3O4 composite particles as effective magnetic assisted adsorbents, Colloid. Surf. A 279 (2006) 167–174.
  10. Wu, H. Joo, K. Lee, Kinetics and thermodynamics of the organic dye adsorption on the mesoporous hybrid xerogel, Chem. Eng. J. 112 (2005) 227–236.
  11. -X. Li, S.-J. Cai, F.-Y. Zheng, Self assembled TiO2 with 5-sulfosalicylic acid for improvement its surface properties and photodegradation activity of dye, Dyes Pigm. 95 (2012) 188–193.
  12. Li, Q. Yue, H. Sun, Y. Su, B. Gao, A comparative study on the properties, mechanisms and process designs for the adsorption of non-ionic or  anionic dyes onto cationic–polymer/bentonite, J. Environ. Manage. 91 (2010)  1601–  1611.

The following is included in the manuscript

Increase in temperature increases the solubility of dye and hence adsorption increases. Increased adsorption may also be as a result of increase in the mobility of the dye ion with temperature. An increasing number of molecules may acquire sufficient energy to undergo an interaction with active site at the surface. Therefore increase in sorption uptake of dyes with increase in temperature may be partly attributed to chemisorption. 

Round 2

Reviewer 1 Report

The authors made all the requested changes. Now the writing is fine and can be accepted in its current state.

Reviewer 2 Report

It's OK